# Effect of *Moringa oleifera* Leaf Extract on the Growth Performance, Hematology, Innate Immunity, and Disease Resistance of Nile Tilapia (*Oreochromis niloticus*) against *Streptococcus agalactiae* Biotype 2

**DOI:** 10.3390/ani14060953

**Published:** 2024-03-19

**Authors:** Manoj Tukaram Kamble, Wenresti Gallardo, Krishna Rugmini Salin, Suntree Pumpuang, Balasaheb Ramdas Chavan, Ram Chandra Bhujel, Seema Vijay Medhe, Aikkarach Kettawan, Kim D. Thompson, Nopadon Pirarat

**Affiliations:** 1Center of Excellence in Wildlife, Exotic, and Aquatic Animal Pathology, Faculty of Veterinary Science, Chulalongkorn University, Bangkok 10330, Thailand; maav.manya@gmail.com (M.T.K.); seemamedhe@gmail.com (S.V.M.); 2Aquaculture and Aquatic Resources Management (AARM), Department of Food, Agriculture and Bioresources, School of Environment Resources and Development, Asian Institute of Technology, Pathum Thani 12120, Thailand; salinkr@ait.ac.th (K.R.S.); suntree@ait.ac.th (S.P.); bhujel@ait.ac.th (R.C.B.); 3Department of Marine Science and Fisheries, College of Agricultural and Marine Sciences, Sultan Qaboos University, Muscat 123, Oman; gallardo@squ.edu.om; 4Department of Aquaculture, College of Fisheries, Ratnagiri 415629, India; brc15672@gmail.com; 5Department of Food Chemistry, Institute of Nutrition, Mahidol University, Nakhon Pathom 73170, Thailand; aikkarach.ket@mahidol.ac.th; 6Moredun Research Institute, Pentlands Science Park, Penicuik EH26 0PZ, UK; kim.thompson@moredun.ac.uk

**Keywords:** *Moringa oleifera*, Nile tilapia, growth performance, hematology, immune response, *Streptococcus agalactiae* Biotype 2

## Abstract

**Simple Summary:**

*Streptococcus agalactiae* Biotype 2 affects Nile tilapia during the grow-out phase, exhibiting a higher virulence with more than 90% mortality compared to Biotype 1, ultimately leading to considerable economic losses in aquaculture. While other therapies are available to treat this bacterium, research on an herbal treatment for a resistance to *S. agalactiae* Biotype 2 infection in Nile tilapia is limited. In this study, we investigated the effects of *Moringa oleifera* leaf (MLE) extract-supplemented diets on the growth, feed utilization, hematology, innate immune response, and disease resistance of Nile tilapia against *S. agalactiae* Biotype 2. Results revealed that the diets supplemented with the MLE-0.5 group showed a significantly higher growth, feed utilization, hematology, innate immune response, and relative percent of survival of Nile tilapia against *S. agalactiae* Biotype 2 compared to the control, MLE-1, MLE-1.5, and MLE-2. In conclusion, MLE-0.5 can be employed as an alternative feed supplement in sustainable Nile tilapia farming to protect against *S. agalactiae* Biotype 2.

**Abstract:**

The present study aimed to investigate the effects of *Moringa oleifera* leaf (MLE) extract-supplemented diets on the growth, feed utilization, hematology, innate immune response, and disease resistance of Nile tilapia against *Streptococcus agalactiae* Biotype 2. Four hundred and fifty Nile tilapia (32.61 ± 0.2 g/fish) were randomly allocated into fifteen tanks (30 fish/tank). Different concentrations of MLE at 0%, 0.5%, 1%, 1.5%, and 2% were fed to the Nile tilapia for 30 days, and the growth, feed utilization, hematology, and innate immune response of the Nile tilapia were determined. After the feeding trial, the Nile tilapia were challenged with a *S. agalactiae* Biotype 2 infection, and the relative percentage of survival (RPS) was determined. Results revealed the presence of quercetin, kaempferol, and *p*-coumaric acid in the MLE extract, exhibiting stronger antimicrobial activity against *S. agalactiae* Biotype 2. The diets supplemented with the MLE-0.5 group showed a significantly higher growth, feed utilization, hematology, and innate immune response in the Nile tilapia compared to the control and other MLE groups. Additionally, the MLE-0.5 group exhibited a significantly higher RPS of the Nile tilapia against *S. agalactiae* Biotype 2. Therefore, MLE-0.5 can be employed as an alternative feed supplement in sustainable Nile tilapia farming to protect against *S. agalactiae* Biotype 2.

## 1. Introduction

Nile Tilapia (*Oreochromis niloticus*) emerges as the prospective species for aquaculture due to advancements in hybridization and genetic engineering techniques, empowering its cultivation across freshwater, brackish, and marine environments [1]. The amount of largest tilapia production occurs in China, followed by Indonesia, Egypt, Brazil, and Thailand [2]. Currently, Nile tilapia ranks third in the major aquaculture species, with 4.4 million tonnes of production in 2020 [2].

The growing demand for animal protein necessitates intensive culture systems, which, unfortunately, elevate disease risk due to their suboptimal water quality, high stocking density, and abrupt shifts in culture conditions. These conditions amplify stress, triggering various disease agents, primarily from streptococcal bacteria [3,4]. Streptococcosis is a common disease observed in various aquaculture species [5]. *Streptococcus agalactiae*, characterized by its phenotypic traits as Biotype 1 (β-haemolytic) and Biotype 2 (nonhaemolytic), significantly contributes to tilapia culture mortality globally [6,7]. The bacterium of Biotype 1 targets juvenile fish, while Biotype 2 affects fish during the grow-out phase, exhibiting a higher virulence with more than 90% mortality compared to Biotype 1 [8,9,10], ultimately leading to considerable economic losses in aquaculture.

Aquaculturists have developed economical and nutritious fish diets incorporating immunostimulants to ensure optimal health during the grow-out phase [11]. Herbal immunostimulants offer a promising alternative to antibiotics, not solely due to their cost-effective and environmentally sustainable antimicrobial attributes but also owing to their capacity to enhance growth performance, accompanied by minimal adverse effects [12,13].

Moringa (*Moringa oleifera*) is renowned for its nutritional value, providing essential fibers, proteins, vitamins, minerals, and lipids [14]. Moringa exhibits anticancer, antidiabetic, anti-inflammatory, antioxidant, antifungal, and antibacterial properties attributed to the presence of bioactive compounds such as phenolic acid and flavonoids [15,16,17,18]. Moringa leaf extract has been employed in various studies to promote the growth, immune response, antioxidant activity, and disease protection of Nile tilapia at different life stages, including fry [19,20,21], fingerling [22,23], and adult [24,25] Nile tilapia. Additionally, a few studies have demonstrated the potential of Moringa leaf for reducing stress indices [20,24], as well as mitigating the sub-lethal toxicity of fipronil [26] and sub-chronic sodium fluoride [27] to Nile tilapia. Furthermore, supplementation with Moringa leaf at 10–20% for carnivorous fish and 10–30% for herbivorous and omnivorous fish shows potential as an alternative feed for aquaculture without any adverse effects [28]. However, there is a need for more research on an in vivo herbal treatment for a resistance to *S. agalactiae* Biotype 2 infection in Nile tilapia. Therefore, this study aimed to investigate the dietary supplementation of *M. oleifera* leaf aqueous extract on the growth and feed utilization performance, hematology, innate immune response, and disease resistance of Nile tilapia against *S. agalactiae* Biotype 2 infection.

## 2. Materials and Methods

### 2.1. Preparation of Moringa Leaf Aqeous Extract

The Moringa leaf was obtained from All-Season Herbs Pvt. Ltd. in Bangalore, India, and an aqueous extract was prepared according to the previously described method [29]. The leaves underwent oven drying at 50 °C for 96 h and were subsequently pulverized. A leaf powder (10 g) was then dissolved and extracted in 100 mL of distilled water, and the mixture was homogenized using an orbital shaker (D-500, Hsiangtai, New Taipei City, Taiwan) at 0.28× *g* for 20 h at room temperature (RT). The resulting mixture was centrifuged at 8000× *g* for 15 min at RT. The supernatant obtained was collected, evaporated using a rotavapor (R-200/205, BÜCHI, Flawil, Switzerland) at 35 °C, and further desiccated in a hot air oven at 50 °C for 48 h. The resultant dry samples were stored in bottles and refrigerated at 4 °C until further use.

### 2.2. Identification of Phenolic Acids and Flavonoid Content in MLE Using High-Performance Liquid Chromatography with a Diode-Array Detection (HPLC-DAD) Analysis

The phenolic acids’ and flavonoids’ contents were analyzed using HPLC-DAD. Leaf extract was prepared using acid hydrolysis [30]. Briefly, samples (0.2 g) were combined with 62.5% methanol containing 0.5 g/L of tert-Butylhydroquinone (tBHQ, 3.2 mL) and 6 N of hydrochloric acid (HCl, 0.8 mL). The mixture was heated at 95 °C for 2 h and subsequently cooled on ice for 5 min, followed by the addition of 20 µL of 1% ascorbic acid. The resultant mixture was transferred to a volumetric flask and adjusted to 5 mL before undergoing sonication for 5 min. Further, it was filtered through a 0.2 µm polytetrafluoroethylene (PTFE) syringe filter for subsequent HPLC analysis. The HPLC-DAD analysis of phenolic acids and flavonoids in MLE was carried out using an Agilent 1260 Infinity II system (Agilent Technologies, Waldbronn, Germany), following the method detailed by Judprasong et al. [31] with modifications. Chromatographic separation was achieved using a 5 μm Zorbax Eclipse XDB-C18 column (4.6 mm × 150 mm, Agilent Technologies, Santa Clara, CA, USA). The mobile phase consisted of a gradient of HPLC water containing 0.05% (*w*/*w*) trifluoroacetic acid (TFA), methanol containing 0.05% (*w*/*w*) TFA, and acetonitrile containing 0.05% (*w*/*w*) TFA, at a constant flow rate of 0.6 mL/min. The column temperature was set at 30 °C, and the sample injection volume was 10 µL. Chromatograms were monitored at 280 (gallic acid), 290 (naringenin and hesperitin), 325 (caffeic acid, *p*-coumaric acid, ferulic acid, and sinapic acid), 340 (luteolin and apigenin), and 370 (myricetin, quercetin, and kaempferol) nm using Agilent’s OpenLab Software (Version C.01.07 SR3, Agilent Technologies, Santa Clara, CA, USA). Peaks were identified by comparing unknown peak retention times and UV spectra to authentic standards. The content of flavonoids and phenolic acids was expressed as µg/100 g of the sample.

### 2.3. Pathogen

The *Streptococcus agalactiae* Biotype 2 was kindly provided by the National Center of Streptococcus Collection, Department of Microbiology, Faculty of Medical Science, Chulalongkorn University, Thailand. It was subsequently cultured in Brain Heart Infusion (BHI) broth at 35 °C overnight. The bacterial stock culture was then preserved in a solution containing 20% glycerol and 0.85% saline at −20 °C until needed.

### 2.4. Antimicrobial Activity of MLE

The antimicrobial activity of MLE was investigated using a disk diffusion assay [32]. An amount of 100 µL of a previously prepared bacterial suspension (adjusted to 0.5 McFarland standards, equivalent to 1 × 10^7^ CFU/mL) was inoculated onto a BHI agar (HI-MEDIA, Kennett Square, PA, USA) surface. Subsequently, a sterile paper disk with a diameter of 6 mm was placed on the inoculated agar plate and loaded with 40 µL of the MLE (stock culture at 100 mg/mL). The DMSO (MERCK, Darmstadt, Germany) was subjected to the same procedure as the control. The plates were incubated at 35 ± 2 °C for 24 h, and the entire process was conducted in triplicate. The assessment of antimicrobial activity involved measuring the diameter of the zone surrounding the well where bacterial growth inhibition occurred.

### 2.5. Minimum Inhibitory Concentraion (MIC) and Minimum Bactericidal Concentraion (MBC) of MLE

The determination of the MIC and MBC for MLE was carried out using the micro-dilution method [32]. The MLE was subjected to two-fold serial dilutions resulting in 12 concentrations (5000, 2500, 1250, 625, 312.5, 156.25, 78.12, 39.06, 19.53, 9.76, 4.88, and 2.44 µg/mL) for both MIC and MBC assessments. A sterile microplate was prepared with 100 µL of BHI, 20 µL of bacterial suspension (1 × 10^7^ CFU/mL) and 100 µL of MLE. The DMSO (MERCK, Darmstadt, Germany) served as the control. The plates were then incubated at 35 ± 2 °C for 24 h, and the entire procedure was conducted in triplicate. Following incubation, the MIC was determined as the lowest concentration of MLE that exhibited no bacterial growth, while the MBC was defined as the lowest MLE concentration with no observable growth in the culture. The antibacterial activity was assessed by calculating the MBC/MIC ratio. A ratio of MBC/MIC ≤ 4 indicated a bactericidal effect, whereas a ratio of MBC/MIC ≥ 4 signified a bacteriostatic effect [33].

### 2.6. Fish

Nile tilapia (32.61 ± 0.2 g) of the Chitralada strain were sourced from the AIT experimental facility. Before initiating the experiment, the fish underwent a 15-day acclimatization period in three 500 L fiberglass tanks. During this period, they were fed pelleted feed twice a day at a rate of 4% of their body weight. Following acclimatization, the fish were randomly allocated to fifteen 150 L glass aquaria, with each aquarium containing 30 fish, all incorporated into a recirculation system.

### 2.7. Preparation of the Experimental Diets Supplemented with MLE

The MLE was dissolved in distilled water and mixed with commercial tilapia feed (Charoen Pokphand (CP-7710), Foods Public Company Limited, Samut Sakhon, Thailand) at a rate of 1 mL per gram of feed to prepare the experimental diets. The resulting mixture underwent a mincing process using a mincer (YC80B-4, MITSUYAMA, Bang Kho Laem, Bangkok, Thailand) to form spaghetti-like strands. Subsequently, the diet was reshaped into pellets approximately 5 mm in length, followed by 24 h of drying at 50 °C in a hot air oven. The dried pellets were then stored at 4 °C (RS265TDWP, Samsung, Suwon, Gyeonggi-do, Republic of Korea) until the end of the experiment.

### 2.8. Experimental Design

The experiment employed a completely randomized design, consisting of five experimental treatments labeled as Control, T_1_, T_2_, T_3_, and T_4_, each in triplicate. The control group was without MLE, whereas the treatment groups (T_1_, T_2_, T_3_, and T_4_) were administered with feed containing 0.5%, 1.0%, 1.5%, and 2% MLE, respectively. The proximate composition of the experimental diets is shown in Table 1.

The fish were fed twice daily at a rate of 5% of their body weight. Water quality parameters, including temperature (30.15 ± 0.06 °C), pH (7.24 ± 0.04), dissolved oxygen (5.20 ± 0.04 mg/L), ammonia nitrogen (0.13 ± 0.01 mg/L), nitrite nitrogen (0.15 ± 0.01 mg/L), and nitrate nitrogen (1.09 ± 0.02 mg/L), were consistently monitored and remained within acceptable ranges throughout the trial. After 30 days of feeding, the growth performance, feed utilization efficiency, hematological and immune parameters were assessed, followed by a bacterial challenge test conducted for an additional 15 days.

### 2.9. Growth Performance and Feed Utilization Efficicency of Nile Tilapia Fed Diets Supplemented with MLE

Following 30 days of the feeding trial, the growth performance and feed utilization efficiency [35] were analyzed using the subsequent equations:Weight gain (WG, g/fish) = (final body weight (FW) − initial body weight (IW))(1)
Specific growth rate (SGR, %/day) = [(ln (FW) − ln (IW)/days] × 100(2)
Survival rate (SR, %) = (No. of fish surviving after feeding trial/No. of fish stocked) × 100(3)
Mean daily intake (MDI, g/fish/day) = ((Total feed intake (FI, g)/T)/No. of fish)(4)
Feed conversion ratio (FCR) = FI (g)/WG(5)
Protein efficiency ratio (PER) = (Wet weight gain (g)/Protein intake (g))(6)

### 2.10. Blood and Serum Collection

The fish underwent 24 h of fasting before the collection of blood samples. Anesthesia was induced using Ethyl 3-aminobenzoate methanesulfonate (MS-222, SIGMA-ALDRICH, St. Louis, MI, USA) at a final concentration of 100 mg/L for blood collection, with fifteen fish randomly selected per treatment. Blood was drawn from the caudal vein using a 1 mL heparinized syringe and promptly transferred to an EDTA tube, gently shaken, and stored in a refrigerator at 4 °C. Additionally, for serum collection, another fifteen fish per treatment were anesthetized, and blood was obtained without the use of EDTA, allowing it to clot for 12 h at room temperature. To separate the serum, tubes were centrifuged in Centrisart^®^ A-14 (Sysmatec, Oberdorfstrasse, Eyholz, Switzerland) at 8000× *g* for 15 min, and the resulting supernatant was transferred to screw-cap Eppendorf tubes and stored at −20 °C for future use.

### 2.11. Hematology Analysis of Nile Tilapia Fed Diets Supplemented with MLE

Hematological parameters such as the red blood cells (RBC), white blood cells (WBC), hematocrit (Hct), hemoglobin (Hb), mean corpuscular volume (MCV), mean corpuscular hemoglobin (MCH), and the mean corpuscular hemoglobin concentration (MCHC) were evaluated using an automatic blood cell counter (HeCo Vet C 9SEAC, Pomezia, Rome, Italy). Hematocrit values were determined using the microhematocrit method and reported as a percentage. For hemoglobin determination, an electrolyte lysis was conducted before the automated analysis.

### 2.12. Innate Immune Paramter Analysis of Nile Tilapia Fed Diets Supplemented with MLE

A turbidimetric method was employed for the quantification of serum lysozyme (LYZ) levels [36]. In 96-well U-bottom microtiter plates, 15 µL of test serum was mixed with 150 µL of *Micrococcus lysodeikticus* (0.3 mg/mL in 0.02 M of a sodium acetate buffer, pH 5.5). The initial optical density (OD) was recorded at 450 nm immediately after adding the substrate, and the final OD was measured after 260 s. The values for serum lysozyme were expressed as Units/mL.

A *S. agalactiae* Biotype 2 suspension was reconstituted at a concentration of 10^7^ CFU/mL in PBS. Subsequently, 100 μL of this suspension was mixed with each serum sample (100 μL) for a duration of 60 min. The mixture was then spread on BHA plates and incubated at 37 °C for 24 h. After incubation, the bacterial colonies were enumerated, and the Serum Bactericidal Assay (SBA) was compared between groups. SBA values were determined using the following equation [37]:SBA (%) = (1 − (No. of colonies after serum treatment/No. of colonies after PBS treatment)) ×100(7)

### 2.13. Disease Resistance of Nile Tilapia Fed Diets Supplemented with MLE against S. agalctiae Biotype 2

After 30 days of feeding, one hundred µL of bacterial suspension (1 × 10^7^ CFU/mL) was intraperitoneally (IP) injected into 30 fish per treatment, and mortality was monitored for an additional 15 days. Throughout the challenge experiment, the average water parameters were maintained at a temperature of 29.89 ± 0.03 °C, pH of 7.49 ± 0.03, and DO of 5.07 ± 0.07 mg/L. To confirm the bacteria was the cause of mortality, tissues were extracted from deceased fish for a bacteriological culture. The cumulative mortality and relative percent of survival (RPS) in the MLE groups were calculated using the following formulas [38]:Cumulative mortality (%) = (mortality after challenge/total number of fish) × 100(8)
RPS (%) = (1 − (mortality (%) in the treated group/mortality (%) in the control group)) × 100(9)

### 2.14. Statistical Analysis

The results were presented in means ± SD. The growth performance, feed utilization, hematology, innate immunity, and mortality data were analyzed using a one-way ANOVA, followed by Duncan’s multiple comparison tests using SPSS version 29 software (SPSS Inc., Chicago, IL, USA). A significance level was set at *p* < 0.05. The cumulative survival percentages of the MLE groups were assessed using the Kaplan–Meier method, and statistical differences were determined through the log-rank (Mantel–Cox) test.

## 3. Results

### 3.1. Phenolic Acid and Flavonoid Content in MLE

The HPLC analysis demonstrated the presence of phenolic acids, such as *p*-coumaric acid and flavonoid contents, including quercetin and kaempferol, in the aqueous extract of the MLE (Figure 1).

The concentration of quercetin (1397.82 ± 138.84 µg/100 g) was higher than that of kaempferol (221.69 ± 25.57 µg/100 g) and *p*-coumaric acid (96.50 ± 3.41 µg/100 g), as shown in Table 2.

### 3.2. Antimicrobial Activity

The MLE exhibited a stronger inhibition zone (17.5 ± 2.3 mm), and the lowest MIC and MBC (78.13 µg/mL) against *S. agalactiae* Biotype 2 compared to the DMSO (Table 3). Additionally, the MLE demonstrated bactericidal activity against *S. agalactiae* Biotype 2.

### 3.3. Improvement of Growth Performance and Feed Utilization Efficiency of Nile Tilapia Fed Diets Supplemented with MLE

The growth performance of the Nile tilapia fed diets supplemented with MLE-0.5 group showed significantly (*p* < 0.05) higher FW, WG, and SGR compared to the control, MLE-1, MLE-1.5, and MLE-2 (Table 4).

The survival rate at the end of the feeding trial did not differ significantly (*p* > 0.05) among the experimental diets. Regarding Feed utilization, the MLE-0.5 group exhibited a significantly (*p* < 0.05) lower FCR and higher PER compared to the control, MLE-1, MLE-1.5, and MLE-2. Additionally, the FI and MDI were not significantly (*p* > 0.05) different among the experimental diets.

### 3.4. Enhancement of the Hematological Parameter of Nile Tilapia Fed Diets Supplemented with MLE

The fish fed the diet supplemented with MLE-0.5 induced a significant (*p* < 0.05) increase in the RBC, WBC, Hb, Hct, MCV, MCH, and MCHC of the Nile tilapia compared to the control and remaining levels of the MLE (Table 5). Furthermore, none of the hematological indices of the MLE-1, 1.5, and 2 were significantly (*p* > 0.05) decreased compared to the control group.

### 3.5. Augmentaion of the Innate Immune Response of Nile Tilapia Fed Diets Supplemented with MLE

The serum lysozyme and bactericidal activity of Nile tilapia were significantly (*p* < 0.05) higher in all MLE-supplemented diet groups compared to the control group (Figure 2). Furthermore, a more heightened lysozyme and bactericidal activity of the Nile tilapia was observed in the MLE-0.5 group than in the remaining MLE groups and the control.

### 3.6. Improvement of the Survival Rate of Nile Tilapia Fed Diets Supplemented with MLE

After a 30-day dietary trial, the fish underwent a challenge with *S. agalactiae* Biotype 2, and the cumulative survival rate was assessed over a 15-day period (Figure 3). The log-rank (Mantel–Cox) test, which evaluated the cumulative survival percentages among all MLE groups, revealed a statistically significant difference (*χ*^2^(4) = 30.930, *p* < 0.01) compared to the control group.

The survival rate (90.0 ± 0.0) and RPS (84.4 ± 2.3) were significantly (*p* < 0.05) higher in MLE-0.5. Furthermore, both the survival rate and RPS decreased dose-dependently with an increasing concentration of the MLE-supplemented diets (Table 6).

During the challenge with *S. agalactiae* Biotype 2, Nile tilapia in the control group exhibited typical clinical signs, including a hemorrhage on the pectoral fin (Figure 4A), body (Figure 4B), bilateral exophthalmia (Figure 4C), a distended abdomen (Figure 4D), corneal opacity (Figure 4E), and pop-eye (Figure 4F).

## 4. Discussion

The *S. agalactiae* Biotype 2 poses a significant threat to fish during the grow-out phase, causing over 90% mortality, resulting in substantial economic losses in aquaculture when compared to Biotype 1 [8,9,10]. However, there is a scarcity of research on herbal remedies to enhance the resistance to *S. agalactiae* Biotype 2 infection in Nile tilapia. This study delves into the impact of MLE extract-supplemented diets on Nile tilapia, exploring aspects such as the Nile tilapia’s growth, feed utilization, hematology, innate immune response, and disease resistance against *S. agalactiae* Biotype 2.

The presence of quercetin, kaempferol, and *p*-coumaric acid was identified in the aqueous extract of moringa leaf, corroborating the findings of moringa leaf extracts from aqueous-acetone [39], and methanol [40,41]. Moreover, the MLE exhibited a stronger inhibition zone and the lowest MIC and MBC and bactericidal activity against *S. agalactiae* Biotype 2. The efficacy of plant extracts in impeding bacterial proliferation has been previously validated for *p*-coumaric acid [42], quercetin [43], and kaempferol [44]. This inhibitory potential likely arises from diverse mechanisms, including alterations in cell membrane permeability, membrane disruption, the interference with DNA gyrase and nucleic acid synthesis, and toxicity resulting from the generation of hydrogen peroxide [42,43,44].

The growth (FW, WG, and SGR) and feed utilization (FCR and PER) performance of Nile tilapia were significantly higher in the diet supplemented with the MLE-0.5 group. Similarly, the diet supplemented with *M. oleifera* leaf extract at 5 g [25] and seed extract at 0.5% [45] observed significantly higher growth indices of Nile tilapia. This phenomenon can be ascribed to the presence of bioactive compounds in MLE, namely *p*-coumaric acid, quercetin, and kaempferol. These compounds have been documented for their ability to enhance the growth of diverse fish species, including common carp (*Cyprinus carpio*) [46], snakehead fish (*Channa argus*) [47], and grass carp (*Ctenopharyngodon idellus*) [48]. Additionally, the augmented growth of Nile tilapia fed a diet supplemented with *M. oleifera* may be attributed to the composition of essential vitamins, minerals, and high-quality protein in *M. oleifera* [49]. Importantly, MLE-1, 1.5, and 2 exhibited no significant reduction in growth and feed utilization compared to the control. This finding is corroborated by previous studies that supplemented *M. oleifera* leaf at 1.5% [20], 5% and 10% [23], which demonstrated no significant enhancement in the performance of Nile tilapia. The insignificant growth observed with higher concentrations of MLE could be attributed to a shorter experimental period and the antinutritional contents such as tannin [50], phytic acid [51,52,53], and saponin [54]. These compounds have been reported to depress or impair the growth performance of fish. Therefore, further studies should be conducted to confirm whether tannin, phytic acid, and saponin are responsible for the insignificant growth performance of Nile tilapia.

The hematopoietic system is regarded as a reflective indicator of the fish’s overall health status, offering insights into its physiological condition [55]. The MLE-0.5 supplemented diet revealed a significant increase in RBC, Hb, Hct, MCV, MCH, and MCHC compared to the control group’s results. Our findings are corroborated by previous studies, which have reported that diets supplemented with different concentrations of *M. oleifera* showed significant enhancement in the RBC, Hb, Hct, MCV, MCH, and MCHC of Nile tilapia [19,22,26,56] and African catfish (*Clarias gariepinus*) [57]. The observed phenomenon can be ascribed to the presence of antioxidants, such as quercetin, in the *M. oleifera* extracts, which are postulated to promote erythropoiesis and mitigate the rate of hemolysis induced by oxidative stress [58]. Importantly, MLE-0.5 exhibited an increase in the WBC count of Nile tilapia, which corroborates the elevation in WBC after diets supplemented with *M. oleifera* extracts in Nile tilapia [22] and African catfish [59]. The increased count of WBCs in the fish could be attributed to the immunomodulatory potential of *M. oleifera* [60]. This augmentation in the WBC count suggests a heightened immunity, which may confer resistance against infections [61].

Lysozyme activity within serum is a pivotal defense mechanism against bacterial pathogens, contributing to a decreased susceptibility to diseases [62]. The detection of pathogens via pattern recognition receptors (PRRs) is vital for triggering innate immune responses and subsequent host defense mechanisms via various signaling pathways, including enhanced lysozyme production, that play a key role in eliminating pathogens [63]. In this study, the serum lysozyme activity was significantly enhanced in all MLE groups compared to the control group. This finding is consistent with previous studies that demonstrated increased LYZ activity in Nile tilapia after dietary supplementation with *M. oleifera* extract [19,25,64] and extracts from *Pisidium guajava* and *Phyllanthus acidus* [4,65]. The diets enriched with MLE are proposed to elevate lysozyme levels, potentially enhancing its effectiveness in targeting the peptidoglycan layer within bacterial cell walls. This action involves the hydrolysis of beta-1,4-glycosidic bonds between N-acetylmuramic acid and N-acetylglucosamine acid [66], ultimately destroying the *S. agalactiae* bacteria. Moreover, serum bactericidal activity represents a recognized mechanism essential for eradicating and eliminating pathogenic microorganisms in fish [67]. Notably, there was a significantly increased SBA in Nile tilapia in all MLE groups compared to the control. This result aligns with previous investigations that reported a significantly increased SBA after dietary supplementation with *Nigella sativa* seed oil in rainbow trout (*Oncorhynchus mykiss*) [68] and *Zingiber officinale* in Asian seabass (*Lates calcarifer*) [69]. The elevation of SBA indicates an increase in the protective proteins in the serum, typically observed following infection challenges. These proteins, referred to as acute phase proteins in inflammation, may increase promptly after infection or injury, affecting hepatic, neuro-endocrine, and immune system functions [70]. The enhancement of the serum LYZ and SBA of Nile tilapia fed diets supplemented with MLE may be correlated with bioactive compounds such as *p*-coumaric acid, quercetin, and kaempferol. These bioactive compounds have demonstrated immunostimulatory properties in common carp [46,71], snakehead fish [47], and zebrafish (*Danio rerio*) [72]. Further research is warranted to elucidate the molecular mechanisms underlying the effects of MLE on Nile tilapia following bacterial infection.

The efficacy of dietary plant extracts can be assessed through challenge studies aimed at enhancing fish resilience against pathogenic infections [73]. Our study demonstrated that diets supplemented with MLE-0.5 exhibited significantly higher survival rates against the *S. agalactiae* Biotype 2 compared to the control. The quercetin-rich MLE extract was found to induce an increase in hematological and immunological responses, potentially playing a crucial role in reducing mortality. This observation aligns with similar effects reported in other studies; for instance, that dietary quercetin reduced the mortality of both Pacific white shrimp (*Litopenaeus vannamei*) to WSSV [74] and of zebrafish to *Aeromonas hydrophila* [72]. Our findings are consistent with previous studies on *M. oleifera* leaf powder, at 1.5 and 5 g/100 g [19], and leaf meal, at 8.58–10.00% [21]. These studies support the effectiveness of *M. oleifera* as a feed supplement for enhancing immunity and controlling the infection caused by *A. hydrophila* and *S. agalactiae* in Nile tilapia. Additionally, Pacific white shrimp showed the highest survival rate when fed a diet containing moringa at 2.5 g per kg against *Vibrio alginolyticus* infection [11].

## 5. Conclusions

In conclusion, the HPLC analysis revealed the presence of *p*-coumaric acid, quercetin, and kaempferol in the MLE aqueous extract, demonstrating stronger antimicrobial activity. Importantly, diets supplemented with MLE-0.5 improved the growth and feed utilization performance, hematology, innate immune response, and protection of Nile tilapia against *S. agalactiae* Biotype 2 infections. Therefore, MLE-0.5 can be employed as an alternative feed supplement in sustainable Nile tilapia farming to protect against *S. agalactiae* Biotype 2 infections.

## Figures and Tables

**Figure 1 animals-14-00953-f001:**
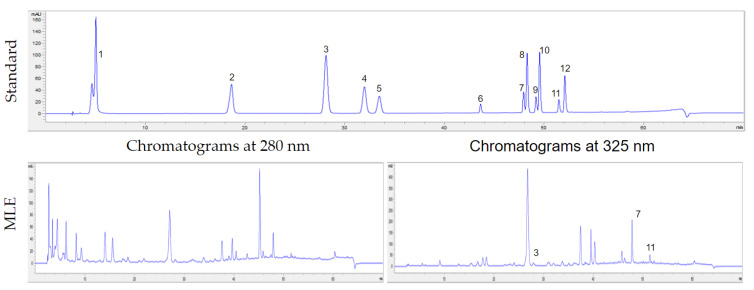
The HPLC chromatograms at 280 nm and 325 nm of standard (phenolic acid and flavonoids), and Moringa leaf extract (MLE). Phenolic acids: 1 = gallic acid; 2 = caffeic acid; 3 = *p*-coumaric acid; 4 = ferulic acid; 5 = sinapic acid. Flavonoids: 6 = myricetin; 7 = quercetin; 8 = luteolin, 9 = naringenin; 10 = hesperitin; 11 = kaempferol; 12 = apigenin.

**Figure 2 animals-14-00953-f002:**
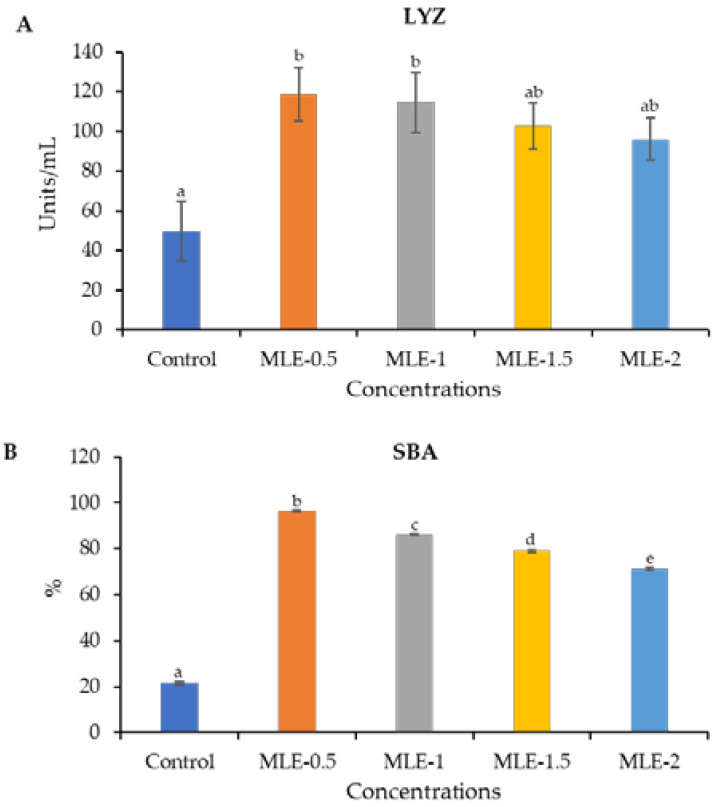
Serum lysozyme activity (LYZ, Units/mL) (**A**) and serum bactericidal activity (SBA, %) (**B**) of Nile tilapia fed diets supplemented with MLE. Data represent the mean ± SD (*n* = 15). Bars assigned with different letters indicate statistical significance (*p* < 0.05).

**Figure 3 animals-14-00953-f003:**
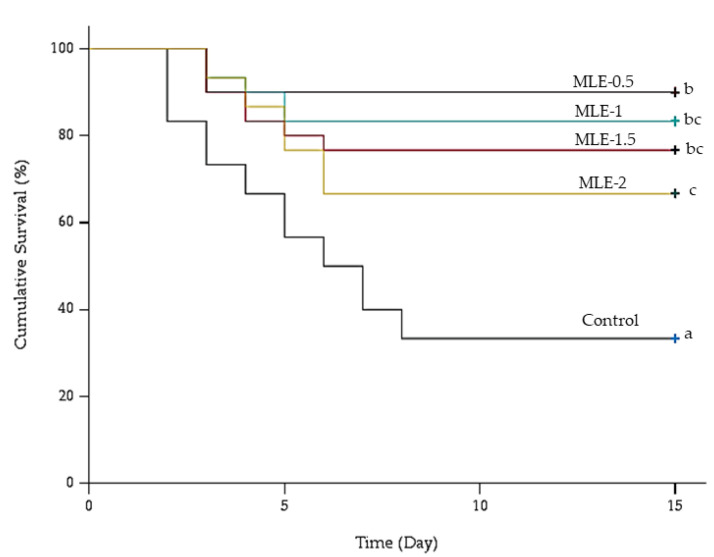
The cumulative survival (%) over time (Day) for Nile tilapia challenged to *S. agalactiae* Biotype 2 (1 × 10^7^ CFU/mL) infection was assessed through Kaplan–Meier curves. Values with means ± SD (*n* = 30 fish/treatment). Different letters indicate statistical significance (*p* < 0.05).

**Figure 4 animals-14-00953-f004:**
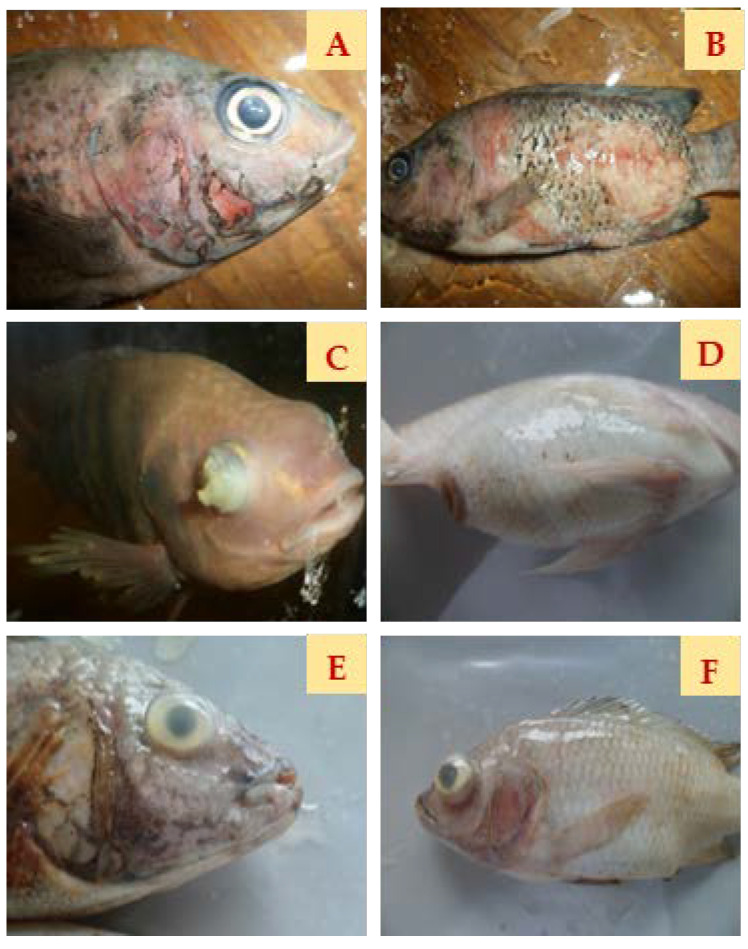
Nile tilapia infected with *Streptococcus agalactiae* Biotype 2 in the control group exhibited clinical signs, including a hemorrhage on the pectoral fin (**A**) and on the body (**B**), bilateral exophthalmia (**C**), a distended abdomen (**D**), corneal opacity (**E**), and pop-eye (**F**).

**Table 1 animals-14-00953-t001:** Proximate composition the diets supplemented with MLE.

Treatments	Crude Protein (%)	Crude Lipid (%)	Moisture (%)	Ash (%)	NFE (%)	GE (Kcal/g)	P/E Ratio
Control	30.43 ± 0.12	7.02 ± 0.10	8.12 ± 0.13	10.20 ± 0.08	44.22 ± 0.13	419.63 ± 0.15	72.52 ± 0.31
MLE-0.5	32.73 ± 1.26	7.09 ± 0.1	8.15 ± 0.12	10.15 ± 0.02	41.88 ± 1.31	423.60 ± 1.90	77.25 ± 2.63
MLE-1	33.36 ± 1.93	7.04 ± 0.10	8.23 ± 0.06	10.15 ± 0.03	41.22 ± 1.94	424.03 ± 3.15	78.66 ± 3.97
MLE-1.5	35.20 ± 0.73	7.09 ± 0.11	8.08 ± 0.07	10.19 ± 0.09	39.44 ± 0.79	427.60 ± 1.00	82.32 ± 1.51
MLE-2	36.00 ± 0.43	7.04 ± 0.09	8.23 ± 0.06	10.16 ± 0.01	38.58 ± 0.28	438.10 ± 0.90	84.10 ± 0.83

The values with means ± SD. NFE: Nitrogen-free extract calculated by difference = 100 − (Crude protein % + Crude lipid % + ash %). Gross energy (GE) was calculated from the National Research Council [34] as 5.65, 9.45, and 4.1 kcal/g for protein, lipid, and NFE, respectively. P/E ratio: Protein energy ratio.

**Table 2 animals-14-00953-t002:** Phenolic acid and flavonoid contents in the MLE.

MLE	Content	µg/100 g
Phenolic acid contents	Gallic acid	ND
	Caffeic acid	ND
	*p*-Coumaric acid	96.50 ± 3.41
	Ferulic acid	ND
	Sinapic acid	ND
Flavonoid contents	Myricetin	ND
	Quercetin	1397.82 ± 138.84
	Kaempferol	221.69 ± 25.57
	Luteolin	ND
	Agpigenin	ND
	Naringenin	ND
	Hesperitin	ND

ND: not detected.

**Table 3 animals-14-00953-t003:** Antimicrobial activity, minimum inhibitory concentration (µg/mL), minimum bacterial concentration (µg/mL) and bactericidal (+) and bacteriostatic (−) reactions of MLE against *S. agalactiae* Biotype 2.

Parameters	Aqueous Extract	DMSO
Zone of Inhibition (mm)	17.5 ± 2.3	0.0 ± 0.0
MIC (µg/mL)	78.13	ND
MBC (µg/mL)	78.13	ND
Reaction	+	ND

Values are means ± SD. ND: not detected.

**Table 4 animals-14-00953-t004:** Growth performance and feed utilization efficiency of Nile tilapia fed diets supplemented with MLE.

	Growth Performance					Feed Utilization Efficiency			
Treatments	IW (g/fish)	FW (g/fish)	WG (g/fish)	SGR (%/day)	SR (%)	FI (Kg)	MDI (g/fish/day)	FCR	PER
Control	32.2 ± 0.1	53.6 ± 1.2 ^a^	21.4 ± 1.2 ^a^	1.70 ± 0.08 ^a^	93.3 ± 5.8	1.24 ± 0.02	1.37 ± 0.02	1.93 ± 0.09 ^a^	71.3 ± 4.1 ^a^
MLE-0.5	32.4 ± 0.1	61.2 ± 1.2 ^c^	28.8 ± 1.2 ^b^	2.12 ± 0.07 ^b^	92.2 ± 6.9	1.25 ± 0.06	1.39 ± 0.07	1.45 ± 0.03 ^b^	95.9 ± 3.9 ^b^
MLE-1	32.5 ± 0.1	55.3 ± 1.9 ^ab^	22.8 ± 1.9 ^a^	1.77 ± 0.11 ^a^	94.4 ± 5.1	1.24 ± 0.10	1.37 ± 0.11	1.82 ± 0.15 ^a^	75.9 ± 6.2 ^a^
MLE-1.5	32.7 ± 0.2	55.7 ± 2.4 ^ab^	23.0 ± 2.5 ^a^	1.77 ± 0.16 ^a^	94.5 ± 6.9	1.26 ± 0.02	1.40 ± 0.02	1.84 ± 0.19 ^a^	76.6 ± 8.4 ^a^
MLE-2	33.0 ± 0.1	57.6 ± 1.3 ^b^	24.6 ± 1.3 ^a^	1.85 ± 0.08 ^a^	96.7 ± 3.4	1.32 ± 0.04	1.47 ± 0.05	1.80 ± 0.11 ^a^	82.0 ± 4.4 ^a^

Values are means ± SD; The same superscript within the same column indicates no significant difference (*p* > 0.05).

**Table 5 animals-14-00953-t005:** Hematological response of Nile tilapia fed diets supplemented with MLE.

Treatments	Control	MLE-0.5	MLE-1	MLE-1.5	MLE-2
RBC (×10^6^ cells/µL)	1.82 ± 0.23 ^a^	2.22 ± 0.27 ^b^	2.01 ± 0.15 ^ab^	1.77 ± 0.54 ^a^	1.86 ± 0.14 ^a^
WBC (×10^2^ cells/µL)	8.04 ± 0.16 ^ab^	9.83 ± 0.61 ^b^	8.43 ± 0.32 ^ab^	9.05 ± 0.31 ^b^	8.36 ± 0.36 ^a^
Hb (g/dL)	6.51 ± 0.52 ^a^	9.05 ± 0.61 ^b^	7.10 ± 0.72 ^ab^	6.00 ± 2.27 ^a^	6.66 ± 0.46 ^a^
Hct (%)	24.78 ± 4.05 ^a^	30.01 ± 3.66 ^b^	27.96 ± 2.52 ^a^	26.38 ± 9.87 ^a^	28.61 ± 2.09 ^ab^
MCV (fl)	138.6 ± 2.73 ^a^	154.2 ± 5.08 ^b^	146.1 ± 6.66 ^ab^	149 ± 8.58 ^ab^	146.6 ±12.10 ^ab^
MCH (pg)	32.85 ± 3.61 ^a^	38.15 ± 1.75 ^b^	35.63 ± 0.75 ^ab^	35.16 ± 0.96 ^ab^	34.8 ± 0.85 ^ab^
MCHC (%)	24.06 ± 1.87 ^abc^	26.18 ± 2.14 ^c^	25.4 ± 0.30 ^bc^	22.4 ± 1.13 ^a^	22.61 ± 1.31 ^ab^

Values with mean ± SD (*n* = 15). The same superscript within the same row indicates no significant difference (*p* > 0.05).

**Table 6 animals-14-00953-t006:** Survival rate (%) and relative percent of survival (%) of Nile tilapia challenged to *S. agalactiae* Biotype 2 after diets supplemented with MLE.

Treatments	Mortality Rate (%)	Survival Rate (%)	RPS (%)
Control	66.7 ± 8.8 ^a^	33.3 ± 8.8 ^a^	-
MLE-0.5	10.0 ± 0.0 ^b^	90.0 ± 0.0 ^b^	84.4 ± 2.3 ^a^
MLE-1	16.7 ± 6.7 ^bc^	83.3 ± 6.7 ^bc^	74.9 ± 9.1 ^ab^
MLE-1.5	23.3 ± 3.3 ^bc^	76.7 ± 3.3 ^bc^	64.0 ± 5.5 ^bc^
MLE-2	33.3 ± 6.7 ^c^	66.7 ± 6.7 ^c^	51.0 ± 5.0 ^c^

Values with means ± SD (*n* = 30 fish/group). The same superscript within the same column indicates no significant difference (*p* > 0.05).

## Data Availability

Data are contained within the article.

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
