# Peer review of "Effect of Moringa oleifera Leaf Extract on the Growth Performance, Hematology, Innate Immunity, and Disease Resistance of Nile Tilapia (Oreochromis niloticus) against Streptococcus agalactiae Biotype 2"

_animals, 2024, doi:10.3390/ani14060953_

Round 1
Reviewer 1 Report
Comments and Suggestions for Authors
see the attached MS with my comments.
Why only MLE0.5 gave better results in terms of almost all the traits tested but NOT MLE 1, 1.5 and 2? Authors should explain this.

Comments on the Quality of English LanguageVery well written except minor mistakes.
Author Response
Response to Reviewer 1 Comments
Point 1: This is a simple summary, So, do not write in general terms as fish. Write as Nile tilapia.
Response: Thank you for the suggestion. We have replaced “fish” with “Nile tilapia” in Line 23 of the revised manuscript.
Point 2: delete “However” and replace it with “While other therapies are available to treat this bacteria,”
Response: Thank you for the suggestion. We deleted “However” and replaced it with “While other therapies are available to treat this bacterium,” in the revised manuscript.
In the revised manuscript
Lines 25-26: While other therapies are available to treat this bacterium,
Point 3: Higher than what group
Response: Thank you for the excellent observation. MLE-0.5 showed higher results compared to the control, MLE-1, MLE-1.5, and MLE-2, and we have incorporated this correction into the revised manuscript.
In the revised manuscript
Lines 31-32: compared to the control, MLE-1, MLE-1.5, and MLE-2.
Point 4: Add control to as 0%
Response: Thank you for the excellent observation. We incorporated a control of 0% in the different concentrations of MLE in line 39 of the revised manuscript.
Point 5: Later in this sentence it is mentioned as “cultivation across freshwater, brackish, and marine environments”. So delete it.
Response: Thank you for the suggestion. We deleted “freshwater” in line 54 of the revised manuscript.
Point 6: delete “the year”
Response: Thank you for the suggestion. We deleted “the year” in line 59 of the revised manuscript.
Point 7: delete “high stocking density”
Response: Thank you for the suggestion. However, “high stocking density” is also one of the factors contributing to the disease risk. Therefore, with all due respect to the reviewer, we are unable to delete “high stocking density” from the revised manuscript.
Point 8: delete “factors” and replace it with “conditions”
Response: Thank you for the suggestion. We deleted “factors” and replaced it with “conditions” in line 62 of the revised manuscript.
Point 9: alternative to what?
Response: Thank you for the excellent observation. We incorporated “alternative to antibiotics,” in line 72 of the revised manuscript.
Point 10: delete “its” and replace it with “presence of”
Response: Thank you for the suggestion. We deleted “its” and replaced it with “the presence of” in line 79 of the revised manuscript.
Point 11: delete “on Moringa leaf have been used” and replace it with “have used moringa leaf”
Response: Thank you for the suggestion. We deleted “on Moringa leaf have been used” and replaced it with “have demonstrated the potential of moringa leaf” in line 83 of the revised manuscript.
Point 12: delete “fpronil” and replace it with “fipronil”
Response: Thank you for the suggestion. We deleted “fpronil” and replaced it with “fipronil” in line 84 of the revised manuscript.
Point 13: is limited or not available? If limited give reference of the available studies. There are some studies available on type 2.
Kamble, M.T., Gallardo, W., Yakupitiyage, A., Chavan, B.R., Rusydi, R. and Rahma, A., 2014. Antimicrobial activity of bioactive herbal extracts against Streptococcus agalactiae biotype 2. Int. J. Basic Appl. Biol, 2(3), pp.152-155.
Lusiastuti, M., Taukhid, T., Anggi, I. and Caruso, D., 2017. Dry green leaves of Indian almond (Terminalia catappa) to prevent streptococcal infection in juveniles of the Nile tilapia (Oreochromis niloticus). Bulletin of the European Association of Fish Pathologists, 37(3), pp.119-125.
Response: Thank you for the suggestion. Herbal treatment for S. agalactiae infection in Nile tilapia is available; however, there is a scarcity of research on in vivo herbal treatment for resistance to S. agalactiae Biotype 2 infection in Nile tilapia. The above-suggested studies by Kamble et al. (2014) focus on the in vitro antimicrobial activity of different plants extracts, while the study by Lusiastuti et al., (2017) was unable to confirm whether it pertains to Biotype 1 or 2. Therefore, we have not included these studies in the previous draft of the manuscript.
Point 14: seems control had significantly lower protein levels compared to the treatments and MLE 1.5 and 2 has significantly higher protein levels than other three. Also the GE and P/E ratio.
Response: Thank you for the excellent observation. However, the proximate compositions of the experimental diets were not significantly different from the control diet, when using Duncan multiple comparison test. Importantly, the protein content of Moringa oleifera leaf ranges from 16% to 40% (Mila et al., 2021; Moyo et al., 2011); therefore, in the present study, MLE-supplemented diets expected to increase with protein content.
References
Milla, P. G., Peñalver, R., & Nieto, G. (2021). Health benefits of uses and applications of Moringa oleifera in bakery products. Plants, 10(2), 318.
Moyo, B., Masika, P. J., Hugo, A., & Muchenje, V. (2011). Nutritional characterization of Moringa (Moringa oleifera Lam.) leaves. African Journal of Biotechnology, 10(60), 12925-12933.
Point 15: This results should be interpreted cautiously because of the difference in protein and GE levels among the treatments including controls.
Response: Thank you for the excellent observation. The proximate compositions of the supplemented diets were not significantly different among the treatments, including the control. Based on this, we have interpreted results of growth performance and feed utilization efficiency of Nile tilapia fed diets supplemented with MLE.
Point 16: There are several data. Is this means all data? If not, please specify.
Response: Thank you for the question. We have updated the description concerning the data in the revised manuscript.
In the revised manuscript
Lines 236 to 239: The growth performance, feed utilization, hematology, innate immunity, and mortality data were analyzed using a one-way ANOVA, followed by Duncan's multiple comparison tests using SPSS version 29 software (SPSS Inc., Chicago, IL, USA). A significance level was set at p < 0.05.
Point 17: were the data normally distributed and what test were used?
Response: Thank you for the question. All the data, including proximate composition of the experimental diets, growth performance, feed utilization efficiency, hematology, innate immune parameters, and the relative percent survival, exhibited a normal distribution. The Shapiro-Wilk Test was employed due to the small sample size (< 50 samples). The Sig. value of the Shapiro-Wilk Test was greater than 0.05, confirming the normality of our data.
Point 18: FI abbreviation. not mentioned earlier.
Response: Thank you for this observation. We have incorporated the abbreviation “FI” into Formula 4 of the revised manuscript.
Point 19: Why only MLE0.5 gave better results in terms of almost all the traits but NOT MLE 1, 1.5 and 2? Authors should explain this.
Response: Thank you for your excellent observation. In the present study, Nile tilapia were fed MLE-supplemented diets for 30 days, resulting in higher growth performance and feed utilization in the MLC-0.5 group compared to the control, MLE1, MLE1.5, and MLE2 groups. Notably, MLE1, MLE1.5, and MLE2 did not reduce growth compared to the control. However, it's important to note a couple of factors. Firstly, the 30-day feeding period might not be sufficient to draw conclusive remarks on growth performance and feed utilization. Secondly, Moringa leaves contain antinutritional contents such as tannin, phytic acid, and saponin, which have been associated with poor growth performance in fish species. Giner-Chavez (1996) reported that tannin levels from 0.5% to 2.0% can depress growth, with levels above 5.0% being potentially lethal. High dietary phytic acid (2.5%) dramatically depressed growth rates in salmon fish (Richardson et al., 1985), and it has been observed that 0.5%–0.6% can impair the growth of rainbow trout (Spinelli et al., 1983) and common carp (Hossain and Jauncey, 1993). Additionally, saponins have been reported as antinutritional factors contributing to poor growth performance in Nile tilapia (Dongmeza et al., 2006). It's worth noting that in the present study, we have not determined the levels of tannin, phytic acid, and saponin in the Moringa leaves or in the experimental diet. This is an important factor to consider, and we have updated this rationale in the revised manuscript.
In the revised manuscript
Lines 354-360: The insignificant growth observed in higher concentrations of MLE could be attributed to a shorter experimental period and the antinutritional contents such as tannin [50], phytic acid [51-53], and saponin [54]. These compounds have been reported to depress or impair growth performance of fish. Therefore, further studies should be conducted to confirm whether the tannin, phytic acid, and saponin are responsible for the insignificant growth performance of Nile tilapia.
References
- Giner-Chavez, B.I. Condensed tannins in tropical forages. Cornell University, Ithaca, NY, USA, 1996.
- Richardson, N.L.; Higgs, D.A.; Beames, R.M.; McBride, J.R. Influence of dietary calcium, phosphorus, zinc and sodium phytate level on cataract incidence, growth and histopathology in juvenile chinook salmon (Oncorhynchus tshawytscha). The Journal of Nutrition 1985, 115, 553-567.
- Spinelli, J.; Houle, C.R.; Wekell, J.C. The effect of phytates on the growth of rainbow trout (Salmo gairdneri) fed purified diets containing varying quantities of calcium and magnesium. Aquaculture 1983, 30, 71-83.
- Hossain, M.; Jauncey, K. The effects of varying dietary phytic acid, calcium and magnesium levels on the nutrition of common carp, Cyprinus carpio. In Proceedings of the Fish Nutrition in Practice. Proceedings of International Conference, Biarritz, France, June 24-27, 1991, 1993; pp. 705-715.
- Dongmeza, E.; Siddhuraju, P.; Francis, G.; Becker, K. Effects of dehydrated methanol extracts of moringa (Moringa oleifera Lam.) leaves and three of its fractions on growth performance and feed nutrient assimilation in Nile tilapia (Oreochromis niloticus (L.)). Aquaculture 2006, 261, 407-422.
Point 20: One recent good review paper can be used.
Van Doan, H., Soltani, M., Leitão, A., Shafiei, S., Asadi, S., Lymbery, A.J. and Ringø, E., 2022. Streptococcosis a Re-emerging disease in aquaculture: significance and phytotherapy. Animals, 12(18), p.2443.
Response: Thank you for the suggestion. We updated reference as suggested in lines 63-64: “Streptococcosis is a common disease observed in various species of aquaculture [5].”
Reviewer 2 Report
Comments and Suggestions for Authors
The review report with related observations and recommendations is attached.

Author Response
Response to Reviewer 2 Comments
Manuscript ID animals-2888584:
Effect of Moringa oleifera Leaf Extract on the Growth performance, Hematology, Innate immunity, and Disease Resistance of Nile tilapia (Oreochromis niloticus) against Streptococcus agalactiae Biotype 2
This study aims to investigate the effects of Moringa oleifera leaf (MLE) extract-supplemented diets on the growth, feed utilization, hematology, innate immune response, and disease resistance of Nile tilapia against Streptococcus agalactiae Biotype 2. This investigation contributes to completing the data on the dietary supplementation of M. oleifera leaf aqueous extract on the growth and feed utilization performance, hematology, innate immune response, and disease resistance of Nile tilapia against S. agalactiae Biotype 2 infection.
General comments on the manuscript are as follows:
Point 1: Abstract: In the abstract and in the materials and methods you mention an experimental period of 30 days. Please mention 10 bibliographic sources in the literature that contain a similar experimental period in the context and experimental design used. In my opinion, this is an insignificant experimental period, I know that it is a minimum of 45 days, and I have applied 60 days.
Response: Thank you for the comment. In the present study, MLE-supplemented diets have been utilized as an alternative feed supplement in sustainable Nile tilapia farming to protect against S. agalactiae Biotype 2. We aim to determine whether a 30-day feeding period can provide higher protection against the pathogen. Concurrently, we have assessed growth, hematology, and immune response to evaluate the efficacy of MLE before challenge. We concur with the reviewer that the experimental duration should exceed 45 days. However, it is crucial to note that this duration is pivotal to elucidate the effects of herbal extract on growth performance and feed utilization efficiency.
Below, we have provided 11 bibliographic sources from the literature that encompass a similar experimental period and experimental design as utilized in this context.
Several studies have administered various herbal extract-supplemented diets for 28 days, including Achyranthes aspera seed extract (Rao and Chakrabarti, 2004, 2005a, 2005b; Rao et al., 2004); bulb (Fall and Ndong, 2007) and plant powder of Allium sativum (Fazlolahzadech et al., 2011); ethanolic leaf extract of Rauvolfia tetraphylla (Yogeshwari et al., 2015); and Echinacea purpurea extract (Rahman et al., 2018). Additionally for a duration of 30 days, studies have included Aegle marmelos Leaves extract (Pratheepa et al., 2011); Artemisia cina plant powder (Saleh et al., 2010); Hygrophila auriculata (Kumar et al., 2022) to evaluate growth, hematology, immune response, and disease resistance of various fish species.
References
Fall, J., & Ndong, D. (2007). The effect of garlic (Allium sativum) on growth and immune responses of hybrid tilapia (Oreochromis niloticus x Oreochromis aureus). Document Scientifique du CRODT, 1-22.
Fazlolahzadeh, F., Keramati, K., Nazifi, S., Shirian, S., & Seifi, S. (2011). Effect of garlic (Allium sativum) on hematological Parameters and plasma activities of ALT and AST of Rainbow trout in temperature stress. Australian Journal of Basic & Applied Sciences 5,84–90.
Kumar, J., Priyadharshini, M., Madhavi, M., Begum, S. S., Ali, A. J., Musthafa, M. S., & Faggio, C. (2022). Impact of Hygrophila auriculata supplementary diets on the growth, survival, biochemical and haematological parameters in fingerlings of freshwater fish Cirrhinus mrigala (Hamilton, 1822). Comparative Biochemistry and Physiology Part A: Molecular & Integrative Physiology, 263, 111097.
Pratheepa, V., Madasamy, D., & Sukumaran, N. (2011). Immunomodulatory activity of Aegle marmelos in freshwater fish (Catla catla) by non-specific protection. Pharmaceutical Biology, 49(1), 73-77.
Rahman, A. N. A., Khalil, A. A., Abdallah, H. M., & ElHady, M. (2018). The effects of the dietary supplementation of Echinacea purpurea extract and/or vitamin C on the intestinal histomorphology, phagocytic activity, and gene expression of the Nile tilapia. Fish & Shellfish Immunology, 82, 312-318.
Rao, Y. V., & Chakrabarti, R. (2004). Enhanced anti-proteases in Labeo rohita fed with diet containing herbal ingredients. Indian Journal of Clinical Biochemistry, 19, 132-134.
Rao, Y. V., & Chakrabarti, R. (2005). Dietary incorporation of Achyranthes aspera seed influences the immunity of common carp Cyprinus carpio. Indian Journal of Animal Sciences, 75(9), 1097.
Rao, Y. V., & Chakrabarti, R. (2005). Stimulation of immunity in Indian major carp Catla catla with herbal feed ingredients. Fish & Shellfish Immunology, 18(4), 327-334.
Rao, Y. V., Romesh, M., Singh, A., & Chakrabarti, R. (2004). Potentiation of antibody production in Indian major carp Labeo rohita, rohu, by Achyranthes aspera as a herbal feed ingredient. Aquaculture, 238(1-4), 67-73.
Saleh, O. A., Sakr, S. F., & Abdelhadi, Y. M. (2010). Effect of wormseed plants; Artemisia cina L. and chamomile; Matricaria chamomilla L. on non specific immune response of Clarias gariepinus (African catfish). Abbassa Int. J. Aqua, 3, 246-259.
Yogeshwari, Govintharaj, Chandrasekar Jagruthi, Sannasi Muthu Anbazahan, Lourthu Samy Shanthi Mari, Jaganathan Selvanathan, Jesu Arockiaraj, Nagarajan Balachandran Dhayanithi, Thipramalai Thankappan Ajithkumar, Chellam Balasundaram, and Harikrishnan Ramasamy. "Herbal supplementation diet on immune response in Labeo rohita against Aphanomyces invadans." Aquaculture 437 (2015): 351-359.
Point 2: Introduction: I recommend improving the introduction chapter by adding more recent references on the subject.
Response: Thank you for the comment. We have added the recent reference on “Streptococcosis” and “Moringa oleifera” in the revised manuscript.
In the revised manuscript
Lines 63 to 64: Streptococcosis is a common disease observed in various species of aquaculture [5].
Lines 85-87: Furthermore, supplementation with Moringa leaf at 10-20% for carnivorous fish and 10-30% for herbivorous and omnivorous fish shows potential as an alternative feed for aquaculture without any adverse effects [28].
References
- Van Doan, H.; Soltani, M.; Leitão, A.; Shafiei, S.; Asadi, S.; Lymbery, A.J.; Ringø, E. Streptococcosis a Re-emerging disease in aquaculture: significance and phytotherapy. Animals 2022, 12, 2443.
- Momin, M.; Memiş, D. Potential use of the miracle tree (Moringa oleifera) leaves in aquaculture: A recent update. Aquatic Sciences and Engineering 2023, 38, 122-130.
Point 3: Line 79: I think there is a mistake in the name of the chemical compound.
Response: Thank you for the excellent observation, and we apologize for the typing error. The correct name, “fipronil,” of the chemical compound has been included in line 84 of the revised manuscript.
Point 4: Materials and methods: Line 149-154: 2.6. Ethical approval: Ethics approval according to journal instructions must be moved to the Institutional Review Board Statement with the required supporting documentation.
Response: Thank you for the suggestion. We moved the ethics approval to the Institutional Review Board Statement in the revised manuscript.
Point 5: Line 182: The dissolved oxygen content (5.20 ± 0.04 mg/L) is low with implications on the physiology of the organism. It is recommended that this content be between 5-8 mg/L for warm water species.
Response: Thank you for the suggestion. In general, tilapias tolerate low DO concentrations, even down to 0.1 mg/L, but maximum growth is achieved with DO concentrations greater than 3 mg/L. Oxygen is essential for fish growth and survival, affecting fish respiration as well as nitrate and ammonia toxicity. Among tilapia species, the minimum DO requirement is 5 mg/L, and both respiration and feeding activities decrease as the DO concentration decreases (Mallya 2007). De Long et al. (2009) reported that DO levels between 5 and 7 mg/L are suitable for tilapia tanks. Therefore, in the present study, the DO level in the experimental tank was 5.20±0.04 mg/L, falling within an acceptable range.
References
Abd El-Hack, M. E., El-Saadony, M. T., Nader, M. M., Salem, H. M., El-Tahan, A. M., Soliman, S. M., & Khafaga, A. F. (2022). Effect of environmental factors on growth performance of Nile tilapia (Oreochromis niloticus). International Journal of Biometeorology, 66(11), 2183-2194.
Mallya, Y. J. (2007). The effects of dissolved oxygen on fish growth in aquaculture. The United Nations University Fisheries Training Programme, Final Project. 30 pp
De Long DP, Losordo TM, Rakocy JE (2009) Tank culture of tilapia. Southern Regional aquaculture center (SARC) Publication No. 282. Education and Extension Service Grant. SRAC No. 2007–38500–18470.
Point 6: Line 176: In Table 1 we report the biochemical composition of the supplemented diet and we observed that the protein content is 30.43% in the control and 36% in the last variant. Don't you think that this difference influences the results (FCR, PER)? I think that you should have adjusted this difference by adding some components to make the protein content uniform in all experimental variants. In the present study these being different, it is not possible to analyze and compare the experimental variants with each other, because there is another factor that you have not considered.
Response: Thank you for the excellent observation. However, it is important to note that the proximate compositions of the experimental diets were not significantly different from the control diet, as determined by the Duncan multiple comparison test. This suggests that the supplementation of Moringa oleifera leaf extract (MLE) did not significantly alter the basic nutrient composition of the diets. Importantly, Moringa oleifera leaf is known to contain protein ranging from 16% to 40% (Mila et al., 2021; Moyo et al., 2011). Therefore, the supplementation of MLE in the present study is expected to increase the overall protein content of the diets. This suggests a potential enhancement in the protein value of the diets through the addition of MLE.
Furthermore, it is noteworthy that there were no significant differences in the proximate compositions among the treatments, including the control. This implies that the variations in diet composition did not significantly affect the overall nutrient content across the different experimental groups. Hence, based on the lack of significant differences in proximate composition among the treatments, we did not anticipate that variations in diet composition would have a substantial impact on the growth performance or feed utilization efficiency of the fish.
Additionally, it's important to highlight that adjustments to compensate for differences in proximate composition were not made by adding supplementary components. This decision was made because the current study utilized MLE as a feed supplement to commercial tilapia feed, rather than altering the feed composition itself.
References
Milla, P. G., Peñalver, R., & Nieto, G. (2021). Health benefits of uses and applications of Moringa oleifera in bakery products. Plants, 10(2), 318.
Moyo, B., Masika, P. J., Hugo, A., & Muchenje, V. (2011). Nutritional characterization of Moringa (Moringa oleifera Lam.) leaves. African Journal of Biotechnology, 10(60), 12925-12933.
Point 7: Line 228: Was the resistance challenge test performed under conditions with dissolved oxygen of 5.07 mg/L? These were not optimal conditions, besides S. agalactiae infection and dissolved oxygen content was a stress factor.
Response: Thank you for this observation. In the present study, we conducted the resistance challenge test based on bacterial concentration rather than dissolved oxygen levels. We also monitored temperature and pH and analyzed DO levels to ensure that these water quality parameters did not affect fish mortality. We did not anticipate that the DO level of 5.07 mg/L during the challenge might stress the fish, as De Long et al. (2009) suggested that DO levels between 5 and 7 mg/L are suitable for tilapia tanks. Therefore, we maintained the DO level in the experimental tank at 5.07 mg/L, which falls within this acceptable range.
Point 8: Results: Line 253: From the results obtained and presented in Table 2, it can be seen that the content of quercitin and coumaric acid is significant, and the results on growth performance show that the lowest concentration is the one that stimulates growth and not only. The question is why the higher concentrations of MLE1, MLE1.5, and MLE 2 did not have beneficial results, given that quercitin and coumaric acid are in a significant concentration. How do you explain these results?
Response: Thank you for your excellent observation. In the present study, Nile tilapia were fed MLE-supplemented diets for 30 days, resulting in higher growth performance and feed utilization in the MLC-0.5 group compared to the control, MLE1, MLE1.5, and MLE2 groups. Notably, MLE1, MLE1.5, and MLE2 did not reduce growth compared to the control. However, it's important to note a couple of factors. Firstly, the 30-day feeding period might not be sufficient to draw conclusive remarks on growth performance and feed utilization. Secondly, Moringa leaves contain antinutritional contents such as tannin, phytic acid, and saponin, which have been associated with poor growth performance in fish species. Giner-Chavez (1996) reported that tannin levels from 0.5% to 2.0% can depress growth, with levels above 5.0% being potentially lethal. High dietary phytic acid (2.5%) dramatically depressed growth rates in salmon fish (Richardson et al., 1985), and it has been observed that 0.5%–0.6% can impair the growth of rainbow trout (Spinelli et al., 1983) and common carp (Hossain and Jauncey, 1993). Additionally, saponins have been reported as antinutritional factors contributing to poor growth performance in Nile tilapia (Dongmeza et al., 2006). It's worth noting that in the present study, we have not determined the levels of tannin, phytic acid, and saponin in the Moringa leaves or in the experimental diet. This is an important factor to consider, and we have updated this rationale in the revised manuscript.
In the revised manuscript
Lines 354-359: The insignificant growth observed in higher concentrations of MLE could be attributed to a shorter experimental period and the antinutritional contents such as tannin [50], phytic acid [51-53], and saponin [54]. These compounds have been reported to depress or impair growth performance of fish. Therefore, further studies should be conducted to confirm whether the tannin, phytic acid, and saponin are responsible for the insignificant growth performance of Nile tilapia.
References
- Giner-Chavez, B.I. Condensed tannins in tropical forages. Cornell University, Ithaca, NY, USA, 1996.
- Richardson, N.L.; Higgs, D.A.; Beames, R.M.; McBride, J.R. Influence of dietary calcium, phosphorus, zinc and sodium phytate level on cataract incidence, growth and histopathology in juvenile chinook salmon (Oncorhynchus tshawytscha). The Journal of Nutrition 1985, 115, 553-567.
- Spinelli, J.; Houle, C.R.; Wekell, J.C. The effect of phytates on the growth of rainbow trout (Salmo gairdneri) fed purified diets containing varying quantities of calcium and magnesium. Aquaculture 1983, 30, 71-83.
- Hossain, M.; Jauncey, K. The effects of varying dietary phytic acid, calcium and magnesium levels on the nutrition of common carp, Cyprinus carpio. In Proceedings of the Fish Nutrition in Practice. Proceedings of International Conference, Biarritz, France, June 24-27, 1991, 1993; pp. 705-715.
- Dongmeza, E.; Siddhuraju, P.; Francis, G.; Becker, K. Effects of dehydrated methanol extracts of moringa (Moringa oleifera Lam.) leaves and three of its fractions on growth performance and feed nutrient assimilation in Nile tilapia (Oreochromis niloticus (L.)). Aquaculture 2006, 261, 407-422.
Point 9: Line 281: In Table 4 as a result of statistical processing you have indicators that you have to mention where they belong e.g. WBC (×102 cells/µL) in the control variant 8.04±0.16ab, MLE-0.5 is 9.83±0.61b; MLE-1 is 8.43±0.32ab; MLE1.5 is 9.05±0.31b; MLE-2 is 8.36±0.36a. Specify a or b.
Response: Thank you for your comment. We have already mentioned in the footnotes of the table regarding statistical comparison: “The same superscript within same row indicates no significant difference (p > 0.05).”
Point 10: For the evaluation of the innate immune response the analysis of lysozyme and antibacterial activity is summary without leading to a pertinent conclusion.
Response: Thank you for your comment. We appreciate the importance of comprehensively evaluating the innate immune response. The level or activity of lysozyme is a crucial indicator of fish innate immunity (Franco Montoya et al., 2017; Amphan et al., 2019). Additionally, the antibacterial activity of blood serum acts as a nonspecific response to inhibit the growth of infectious microorganisms (Yano, 1996). While we respect the reviewer’s perspective, we believe that further research is warranted to elucidate the molecular mechanisms underlying the effects of MLE on Nile tilapia following bacterial infection.
In the revised manuscript
Lines 402-403: Further research is warranted to elucidate the molecular mechanisms underlying the effects of MLE on Nile tilapia following bacterial infection.
References
Yano, T. (1996). The nonspecific immune system: humoral defense. The fish immune system: organism, pathogen, and environment, 105-157.
Montoya, L. N. F., Martins, T. P., Gimbo, R. Y., Zanuzzo, F. S., & Urbinati, E. C. (2017). β-Glucan-induced cortisol levels improve the early immune response in matrinxã (Brycon amazonicus). Fish & Shellfish Immunology, 60, 197-204.
Amphan, S., Unajak, S., Printrakoon, C., & Areechon, N. (2019). Feeding-regimen of β-glucan to enhance innate immunity and disease resistance of Nile tilapia, Oreochromis niloticus Linn., against Aeromonas hydrophila and Flavobacterium columnare. Fish & Shellfish Immunology, 87, 120-128.
Point 11: Discussion: The discussions are very vague and do not correlate with the experimental results obtained. They need to be consistently improved, especially on this species which is intensively studied.
Response: Thank you for your comment. We have every effort to revise the discussion where necessary and have incorporated these changes in the revised manuscript. However, we greatly value the input of reviewers, and we are dedicated to enhancing the quality and clarity of our discussions to improve the scientific rigor of our study. If you have any further suggestions or specific areas of concern, kindly share them with us, and we will address them accordingly.
In the revised manuscript
Lines 354-360: The insignificant growth observed in higher concentrations of MLE could be attributed to a shorter experimental period and the antinutritional contents such as tannin [50], phytic acid [51-53], and saponin [54]. These compounds have been reported to depress or impair the growth performance of fish. Therefore, further studies should be conducted to confirm whether tannin, phytic acid, and saponin are responsible for the insignificant growth performance of Nile tilapia.
Lines 376-380: Lysozyme activity within serum is a pivotal defense mechanism against bacterial pathogens, contributing to decreased susceptibility to diseases [62]. The detection of pathogens via pattern recognition receptors (PRRs) is vital for triggering innate immune responses and subsequent host defense mechanisms via various signaling pathways, including enhanced lysozyme production, play a key role in eliminating pathogens [63].
Lines 384-388: The diets enriched with MLE is proposed to elevate lysozyme levels, potentially enhancing its effectiveness in targeting the peptidoglycan layer within bacterial cell walls. This action involves the hydrolysis of beta-1,4-glycosidic bonds between N-acetylmuramic acid and N-acetylglucosamine acid [66], ultimately destroying the S. agalactiae bacteria. Moreover, serum
Lines 394-398: The elevation of SBA indicates an increase in protective proteins in the serum, typically observed following infection challenges. These proteins, referred to as acute phase proteins in inflammation, may increase promptly after infection or injury, affecting hepatic, neuro-endocrine, and immune system functions [70].
Lines 402-403: Further research is warranted to elucidate the molecular mechanisms underlying the effects of MLE on Nile tilapia following bacterial infection.
Point 12: Conclusions: The conclusions are reported based on results that are not obtained in the appropriate experimental design.
Response: Thank you for your comment. Regarding the experimental period clarification, MLE-supplemented diets were utilized in our study as an alternative feed supplement in sustainable Nile tilapia farming, aiming to protect against S. agalactiae Biotype 2. Our objective was to determine whether a 30-day feeding period could offer greater protection against the pathogen. Additionally, we conducted assessments of growth, hematology, and immune response to evaluate the efficacy of MLE before the challenge. Based on these findings, we included our conclusions in the initial submission of the manuscript.
Point 13: References: References can be improved.
Response: Thank you for your comment. We have improved the references in the revised manuscript.
Reviewer 3 Report
Comments and Suggestions for Authors
The reviewer considers that the manuscript is well written, the objectives are correctly defined, the results are well presented and the conclusions are based on the results obtained. The reviewer's only doubt is the originality and novelty of the research, which is why he recommends that the authors appropriately highlight the differential aspects of their study.
In addition, some small editing errors must be corrected:
Page 3, lines 100 and 104: define abbreviations tBHQ and PTFE.
Page 4, line 152: add the abbreviation of the Asian Institute of Technology (AIT) since it is the first time it is mentioned.
Page 8, line 284: change augumentation for Augmentation.
Page 10, line 323: include the appropriate citation after the first sentence (The S. agalactiae biotype 2....when compared ti biotype 1.
Author Response
Response to Reviewer 3 Comments
The reviewer considers that the manuscript is well written, the objectives are correctly defined, the results are well presented and the conclusions are based on the results obtained. The reviewer's only doubt is the originality and novelty of the research, which is why he recommends that the authors appropriately highlight the differential aspects of their study.
Response: Thank you for your comment. The novelty of our research lies in addressing the limited exploration of herbal treatments for combating Streptococcus agalactiae Biotype 2 infection in Nile tilapia, a significant concern in aquaculture due to its high virulence and economic impact. While other therapies exist, our study focused on investigating the effects of Moringa oleifera leaf (MLE) extract-supplemented diets over a shorter period (30 days) on various aspects of Nile tilapia health and disease resistance against S. agalactiae Biotype 2.
Our findings demonstrated that Nile tilapia fed with diets supplemented with MLE-0.5 exhibited significantly enhanced growth, feed utilization, hematology, innate immune response, and relative percent survival against S. agalactiae Biotype 2 compared to control and other MLE concentration groups. This highlights the potential of MLE-0.5 as a promising alternative feed supplement for sustainable Nile tilapia farming, offering protection against this virulent pathogen.
In summary, our study contributes novel insights by presenting MLE-0.5 as an effective means to enhance Nile tilapia resistance to S. agalactiae Biotype 2 infection, thereby addressing a gap in current research and offering practical implications for aquaculture management.
In addition, some small editing errors must be corrected:
Point 1: Page 3, lines 100 and 104: define abbreviations tBHQ and PTFE.
Response: Thank you for the suggestion. We have incorporated the abbreviations of tBHQ and PTFE in lines 108 and 112 of the revised manuscript.
Point 2: Page 4, line 152: add the abbreviation of the Asian Institute of Technology (AIT) since it is the first time it is mentioned.
Response: Thank you for the suggestion. We have followed your suggestion and incorporated the changes in the revised manuscript.
Page 8, line 284: change augumentation for Augmentation.
Response: Thank you for the suggestion. We have corrected spelling error of Augmentation in the revised manuscript.
Page 10, line 323: include the appropriate citation after the first sentence (The S. agalactiae biotype 2....when compared ti biotype 1.
Response: Thank you for the suggestion. We have followed your suggestion and incorporated the appropriate citation in the sentence.